# Microbial Community Structure and Its Driving Environmental Factors in Black Carp (*Mylopharyngodon piceus*) Aquaculture Pond

Xuemei Li , Lu Liu, Yongjiu Zhu, Tingbing Zhu, Xingbing Wu and Deguo Yang *

Key Laboratory of Freshwater Biodiversity Conservation, Ministry of Agriculture and Rural Affairs of China Yangtze River Fisheries Research Institute, Chinese Academy of Fishery Sciences, Wuhan 430223, China; xmli@yfi.ac.cn (X.L.); liuwanglu528@163.com (L.L.); zhuyj@yfi.ac.cn (Y.Z.); zhutb@yfi.ac.cn (T.Z.); wxbing@yfi.ac.cn (X.W.)

\* Correspondence: yangdg@yfi.ac.cn; Tel.: +86-13995586090; Fax: +86-027-81780108

**Abstract:** This study focused on monitoring the dynamics of bacterial communities and assessment of the influences of physicochemical parameters during a culture cycle in black carp polyculture ponds. Our results showed high variation in microbial diversity and microbial composition in the water column during the culture period. Proteobacteria, Actinobacteria, and Bacteroidetes were the most abundant phyla, while the abundances of the phyla Cyanobacteria, Firmicutes, and Bacteroidetes changed in different months. Moreover, 13 abundant genera with significant difference were found between different months. Thus, samples in different months were divided into three groups according to principal coordinate analysis (PCoA) and unweighted pair-group method (UPGMA) clustering results. RDA showed that total nitrogen (TN), total phosphorus (TP), phosphate ($PO_4^{3-}$-P), nitrate ($NO_3^{-}$-N), temperature (T), dissolved oxygen (DO), and pH significantly shaped the microbial community composition in different months. While Pearson correlation coefficient showed that T, SD, and pH were strongly correlated to the dominant genera. Considering some genera are potential pathogenic bacteria, we could manage the black carp pond by quickly monitoring the water temperature and SD in the future.

**Keywords:** black carp aquaculture pond; microbial community; water column; environmental factors



## 1. Introduction

Aquaculture plays a dominant role in the global economy due to its significant contribution to the food and nutrition supply. It has been reported that Asia accounts for approximately 89% of the world's aquaculture production, with China being the main fish producer and the largest exporter of fish and fish products [1–3] As the major aquaculture method in China, pond culture accounted for 74.0% of the total freshwater aquaculture production of 30,137,441 tonnes in 2019 according to MAFBC (2020) [4]. However, with the rapid growth of pond culture, various aquatic environmental problems such as water pollution and disease outbreaks have occurred [5]. Fish feed may be the main reason for pond water pollution. In order to meet the needs of fish growth, the protein requirement in fish feed is reported to be approximately two to three times higher than that of mammals [6]. However, the conversion efficiencies of feed nitrogen and phosphorus are in the ranges 20–50% and 15–65%, respectively, causing high nitrogen and phosphorus residues in pond water [6,7].

Microorganisms present in the pond sediment and water play crucial roles in nitrogen [8], phosphorus, and carbon cycles [9]. The microbes can remove nitrogen products by nitrification and denitrification. For example, the chemoautotrophic bacteria *Nitrosomonas* and *Nitrobacter* can oxidize ammonium ($NH^{4+}$-N) ions to nitrite ($NO_2^{-}$-N) and nitrate ($NO_3^{-}$-N) ions, which are then assimilated and removed by aquatic plants, algae, and

bacteria as a source of nitrogen [2,10,11]. Intensive pond culture is essentially a process of the input and output of nutrient elements, and nitrogen metabolism is the most crucial microbial process. Indeed, the accumulation of toxic ammonia and nitrite in the water column is the major problem in intensive aquaculture, reducing water quality and leading to fish diseases [12,13]. Therefore, elucidating the microbial composition and identifying the influence factors in the pond water will be useful for controlling the microbial community for both optimal water quality and optimal feeding of fish, ultimately affecting the fishes' health.

The black carp (*Mylopharyngodon piceus*) is a carnivorous freshwater fish of the Cyprinidae family; the species is widely cultivated in China [14]. The fish has also been introduced abroad for its potential to alter food webs by eliminating or reducing snails [15]. The production of black carp was 679,582 tonnes in 2019, comprising > 2.2% of the total freshwater-cultured fish annual output [4]. Meanwhile, the traditional pond polyculture systems are the main mode of fish production. In the studied polyculture systems, the black carp was the main culture species, with a smaller proportion of filter-feeding fish (*Hypophthalmichthys molitrix* and *Aristichthys nobilis*) to maximize the growth potential of black carp, and productivity of the pond. Microorganisms and their potential functions in aquaculture systems have been well studied [16]; however, in order to improve the micro ecological environment quickly by monitoring the water quality, we need to understand the relationships between microorganisms and environmental factors in specific culture modes. Thus, in this study we analyzed the dynamic changes of microbial composition and the main influencing factors during a culture period. Our aims were to reveal the characteristics of the microbial community in black carp pond water and explore the driving environmental factors during the culture period. This valuable information is a fundamental step in understanding biochemical processes in pond water, and is of practical use in management of black carp aquaculture ponds.

## 2. Methods

### 2.1. Experimental Design and Sampling Procedure

In this study, 6 representative aquatic intensive earthen ponds dominated by black carp were sampled for water collection, of which three ponds had large areas (named as BP1, BP2, and BP3) with length 5 m × width 5 m × height 3 m, while the other three ponds (named as SP1, SP2, and SP3) had smaller areas (length 3 m × width 5 m × height 3 m). The ponds were located in central China in Jinan Town, Jingzhou City, Hubei Province. *M. piceus* with mean body weight of 1831.5 ± 106.4 g, *H. molitrix* with mean body weight of 705.48 ± 36.4 g and *A. nobilis* with mean body weight of 193.2 ± 13.4 g were randomly 1000-mL water samples from the two types distributed to the ponds, and their densities were the same and were 469.56 g/m$^2$, 259.56 g/m$^2$, and 31.30 g/m$^2$, respectively in all ponds. Commercially formulated feed (crude protein ≥ 32.0%, crudelipid ≥ 3.0%, lysine ≥ 1.4%, total phosphorus ≥ 0.6%, crude ash ≤ 17.0%, crude fiber ≤ 10.0%, and moisture ≤ 12.0%) was fed to fish during the whole culture period. In each pond, water samples were collected from 2–3 sites (Inlet and outlet or Inlet, outlet and pond center) and then homogeneously pooled for further analyses of water quality and microbial communities. In the middle of each month from May to October in 2015, six of ponds were collected in 0.5 m depth using a plexiglass water collector on sunny days. Each sample was divided into duplicate: 500 mL was used for analyzing water physiochemical parameters, and another 500 mL was used for microbial analysis.

### 2.2. Physicochemical Measurements

The water temperature (T), dissolved oxygen (DO), and pH were measured with a multi-parameter water quality analyzer (HACH-hq40d, HACH, Loveland, CO, USA) at the water surface (0.5 m depth) in situ. The transparency was measured by a Secchi disk. The concentrations of total nitrogen (TN), total phosphorus (TP), $NH_4^+$-N, $NO_3^-$-N, $NO_2^-$-N,

and phosphate ($PO_4^{3-}$-P) were analyzed using a portable multi-parameter spectrophotometer (Hach DR1900, Hach, USA) according to the manufacturer's manual.

### 2.3. Water DNA Extraction and 16S rRNA Gene Amplification

Each 500-mL water sample was filtered by glass-fiber GF/C filters and Millipore filters with 0.45- and 0.22-μm pore sizes, respectively. The two filters that were used for the same water sample were pooled together and kept at −20 °C for DNA extraction. Microbial genomic DNA was extracted using a DNeasy Powerwater Kit (Qiagen, Hilden, NRW, Germany) and purified using a Universal DNA Purification Kit (Tiangen Biotech, Guangzhou, China) according to the manufacturer's instructions. The DNA concentration and quality was assessed using a Nanodrop 2000 and stored at −20 °C for use.

Universal primers 515F (5′-GTGCCAGCMGCCGCGGTAA-3′) and 909R (5′-CCCCG YCAATTCM TTTRAGT-3′) were used to amplify the bacterial 16S rRNA gene V4-V5 fragments. PCR integrant and protocols were carried out as described in [11]: the PCR mixture (25 μL) contained 1× PCR buffer, 1.5 mM $MgCl_2$, each dNTP at 0.4 μM, each primer at 1.0 μM and 0.5 U of ExTaq (TaKaRa, Dalian, China) and 10 ng water genomic DNA. 94 °C for 3 min, followed by 30 cycles of 94 °C for 40 s, 56 °C for 60 s, 72 °C for 60 s, and a final extension at 72 °C for 10 min until the reaction was halted by the user. The PCR products were separated by 2% agarose gel electrophoresis, and negative controls were always performed to make sure there was no contamination. The band with the correct size was excised and purified using a SanPrep DNA Gel Extraction Kit (Sangon Biotech, Shanghai, China) and quantified using a Nanodrop. All samples were pooled, with equal molar amounts from each sample. The sequencing samples were prepared using a TruSeq DNA kit according to the manufacturer's instructions. The purified library was diluted, denatured, re-diluted, mixed with PhiX (equal to 30% of final DNA amount) as described in the Illumina library preparation protocols, and then applied to an Illumina Miseq system for sequencing with the Reagent Kit v2 2 × 250 bp as described in the manufacturer's manual.

### 2.4. Sequencing Data Analysis

QIIME (v1.7.0) was used to process and quality-filter the raw fastq files according to a quality-control process [17]. All sequence reads were trimmed and assigned to each sample based on their barcodes. The sequences with high quality (length > 300 bp, without ambiguous base 'N', and average base quality score > 30) were used for downstream analysis. Chimera sequences were removed using the UCHIME algorithm [18]. The processed sequences with ≥97% similarity were clustered into the same Operational Taxonomic Units (OTUs) by the UCLUST algorithm. Alpha diversity (including Chao, Shannon, and Simpson indices) and beta diversity (including PCoA and UPGMA based on weighted UniFrac distance) were calculated, and analysis of similarities (ANOSIM) was used to verify the difference between groups with vegan package. Taxonomic assignments of each OTU were determined using the RDP classifier [19].

### 2.5. Statistical Analysis

Significant differences in water microbial alpha diversity and environmental variables between different months were determined using a Student's *t*-test or Mann–Whitney rank sum test. The microbial difference on genus levels between months were characterized by Linear Discriminant Analysis Effect Size (LEfSe) analysis through an online toolkit (12-10-2021, http://huttenhower.sph.harvard.edu/galaxy). Redundancy analysis (RDA) was used to calculate the relationship between the water environmental variables and the microbial community at the phylum level due to the eigenvalue < 3 based on the detrended correspondence analysis (DCA). The correlations between environmental variables and microbial communities were analyzed using the Mantel test, and the relationships were visualized in the plots. Pearson correlation coefficient between the dominant genus and environmental variables was computed using corrplot package. Statistical analyses were

performed with the software SPSS 22.0 (IBM, New York, NY, USA) and R (ver. 4.0.3). The level of significance was set at a *p*-value of < 0.05.

### 2.6. Nucleotide Sequence Accession Number

The raw MiSeq pyrosequencing data have been deposited in the NCBI Sequence Read Archive under accession number: PRJNA722102.

## 3. Results

### 3.1. Microbiota Diversity and Richness in the Pond Water Column

After quality filtering and normalization, 983,456 high-quality bacterial sequences were obtained from the 36 water samples, with a mean sequence number of 27,318 per sample. Three richness and diversity indices of bacterial communities were calculated (Figure 1). The results showed higher Shannon and Simpson diversity indices for bacteria in the mid culture period (June, July, and August) in BP ($p < 0.05$), while Chao1 richness showed no differences among months in BP. By comparison, microbiota diversity and richness showed no difference in SP during the culture cycle ($p > 0.05$).

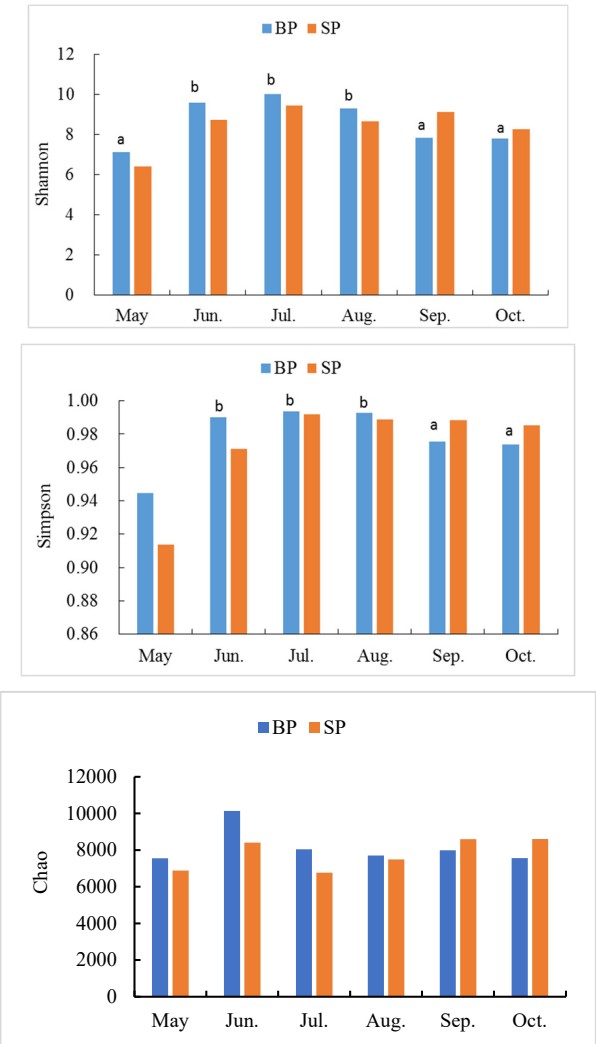

**Figure 1.** Microbial diversity and richness in ponds with large (BP) and small (SP) area during a culture cycle. Letters a and b indicate significant differences ($p < 0.05$) among sampling months.

### 3.2. Microbial Composition and Community Structure in the Pond Water Column

The number of OTUs was analyzed for each sample with a 97% sequence similarity cutoff value. A total of 56,477 OTUs were detected. Among the microbial taxa, the dominant

OTUs across all water samples were classified into 10 phyla (with relative abundance > 1%). The phyla Cyanobacteria, Proteobacteria, Actinobacteria, and Bacteroidetes with relative abundances of 43.7 ± 6.1%, 24.9 ± 1.9%, 12.1 ± 5.3%, and 9 ± 2.1%, respectively, were dominant in samples collected in May. In the pond water collected in June, July, and August, the most abundant taxa were Proteobacteria (42.4 ± 11.6%), Firmicutes (14.6 ± 6.7%), Actinobacteria (12.5 ± 8.3%), and Bacteroidetes (8.8 ± 1.6%). In September and October, Proteobacteria (37.6 ± 2.9%), Actinobacteria (20.1 ± 1.4%), Cyanobacteria (14.0 ± 4.6%), and Bacteroidetes (8.3 ± 2.1%) dominated the pond water (Figure 2).

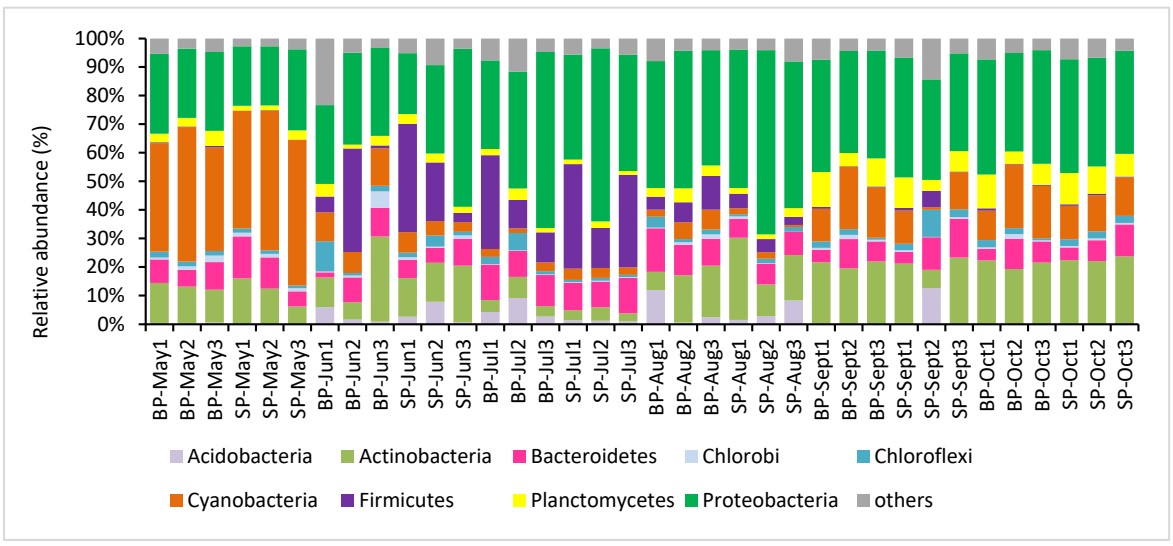

**Figure 2.** Relative abundance of the dominant bacterial phyla for pond water in different months. BP, large area pond; SP, small area pond; May–October indicate sampling time; 1, 2, and 3 represent three duplicate ponds.

LEfSe analyses based on microbial taxa at different genus levels were also performed to illustrate the differences in microbial taxa between different months (Figure 3). Thirteen of the most abundant genera with significant difference were found between different months. In May, *Prochlorococcus* belonging to phylum Cyanobacteria were found to be more abundant. In June, *Bacillus* and *Polynucleobacter* belonging to phyla Firmicutes and Proteobacteria, respectively were dominant. In July, *_Flavobacterium* (phylum Bacteroidetes), *Oscillospira* and *Ruminococcaceae* (phylum Firmicutes), *Agrobacterium*, *Comamonas*, *Janthinobacterium* and *Rheinheimera* (phylum Proteobacteria) were more abundant. While *Chryseobacterium* (phylum Bacteroidetes), *Novosphingobium*, and *Acinetobacter* (phylum Proteobacteria) dominated in August. No more dominant genus were found in September and October.

To investigate the dependence of water microbial community structure in black carp polyculture ponds, PCoA and UPGMA clustering based on weighted UniFrac distance at the OTU level were visualized (Figure 4). The results showed that the microbial communities in the water column from the same sampling month tended to cluster together. All samples were divided into three groups; samples collected in May clustered together, while samples collected in September and October clustered as another group, and samples collected in June, July, and August clustered together as a third group. Samples collected from different area ponds revealed no dispersion. The difference between groups was greater than that within groups according to ANOSIM (R = 0.832, *p* = 0.001).

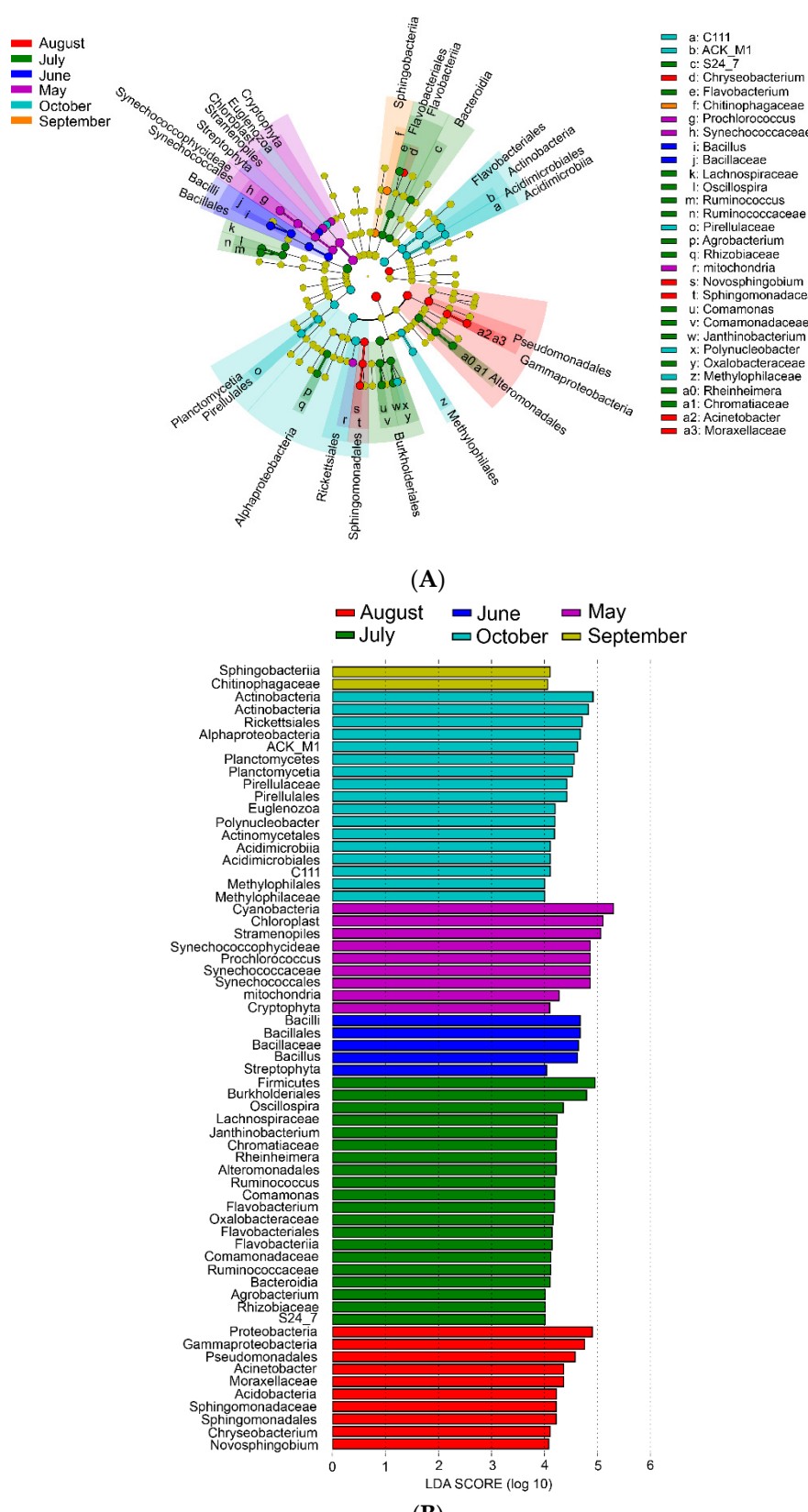

**Figure 3.** LEfSe analysis of different microbial taxa between different months. (**A**) Cladogram representing the LEfSe results of bacteria at different genus levels. Different colors represent different months. (**B**) Histogram of different microbial taxa found by LEfSe ranking between different months. Only taxa with LDA scores > 4.0 are shown.

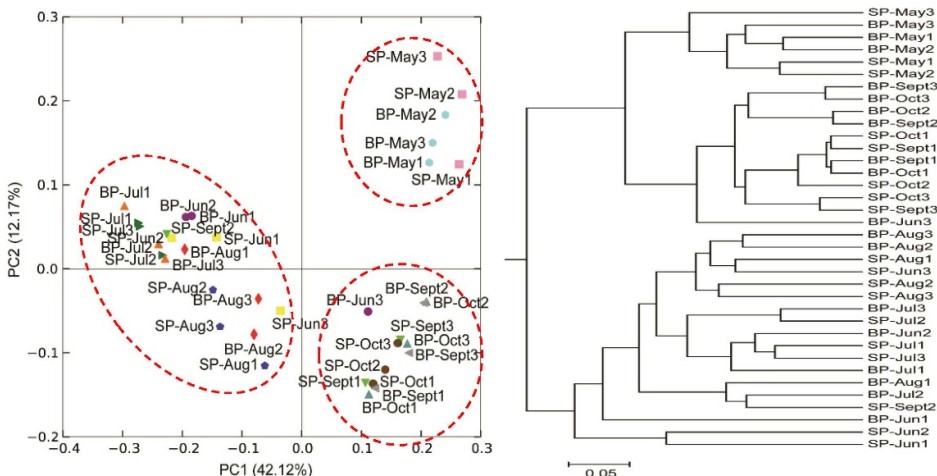

**Figure 4.** PCoA (**left**) and UPGMA (**right**) clustering based on weighted UniFrac distance at the OTU levels. BP, large area pond; SP, small area pond; May–October indicate sampling time; 1, 2, and 3 represent three duplicate ponds.

### 3.3. Key Factors Affecting Microbial Community Composition in the Pond Water Column

Environmental factors in BP and SP ponds during the culture period were showed in Table S1. The factors explaining variation in microbial communities among different sampling times were evaluated by RDA, and the variables in the first two axes collectively explained 58.82% of the variance for the microbial communities among the samples (Figure 5). The results showed that T, pH, DO, TP, TN, $PO_4^{3-}$-P, and $NO_3^-$-N significantly influenced the microbial community in the water column regardless of the size of the pond (Monte Carlo permutation test, $p = 0.01$). The microbial communities sampled in May were mostly affected by TP, while $PO_4^{3-}$-P, pH, and DO affected the microbial communities in September and October, and T and TN were the main factors influencing the microbial communities in June, July, and August.

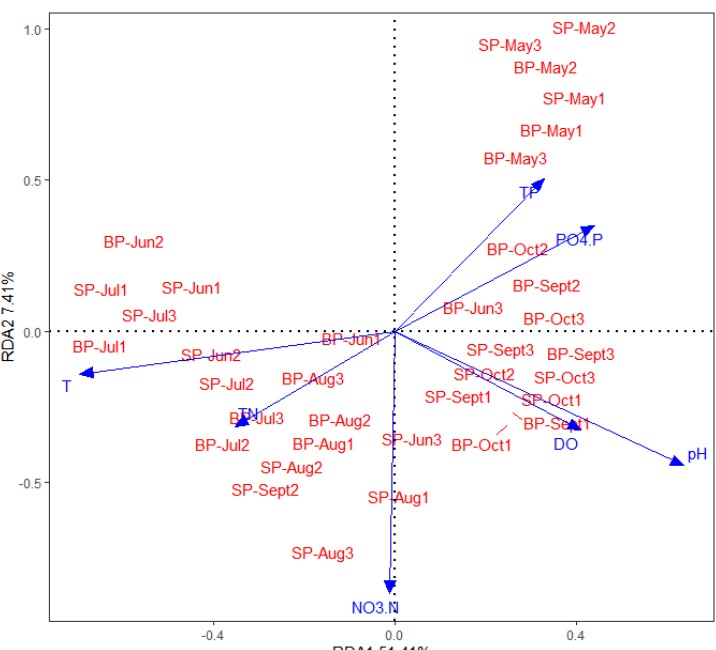

**Figure 5.** RDA diagram of the relationships between pond microbial composition and environmental factors in different sampling months. The explanatory variables are shown by different arrows. BP, large area pond; SP, small area pond; May–October indicate sampling time; 1, 2, and 3 represent three duplicate ponds.

To further elucidate the driving environmental factors on dominant genera in different months, Pearson correlation coefficient was computed and showed in Figure 6. Overall, T was the strongest positive correlates of *Prochlorococcus* and *Chryseobacterium* while the strongest negative correlates of *Acinetobacter*, *Rheinheimera*, *Polynucleobacter*, and *Janthinobacterium*. SD was strongly correlated to *Flavobacterium* and *Rheinheimera*. While pH was strongly negatively correlated to *Polynucleobacter* and *Janthinobacterium* and weakly positively correlated to *Oscillospira* and *Ruminococcaceae*. No significant correlations were found for nutrients except for $NO_3^-$-N, which was negatively correlated to *Polynucleobacter*.

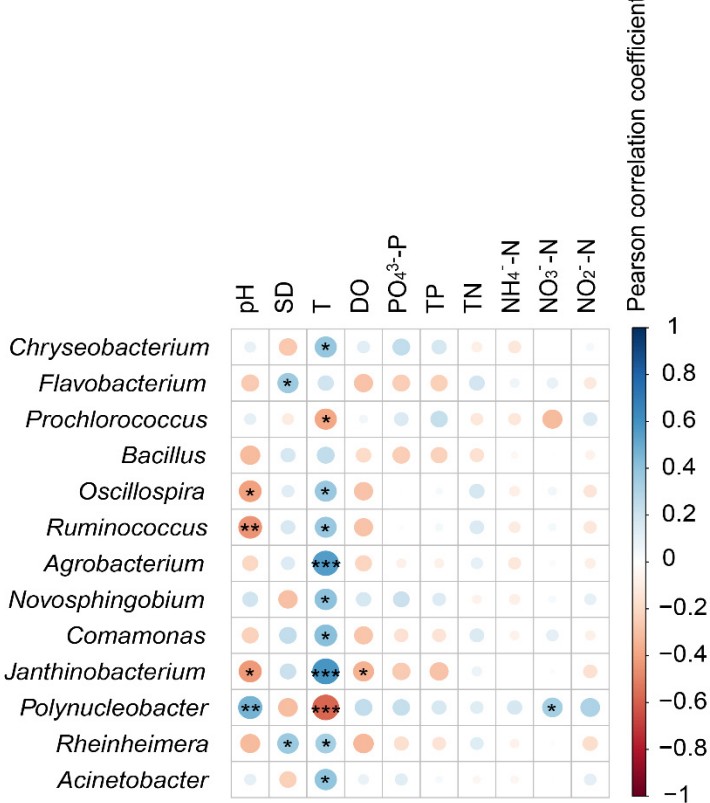

**Figure 6.** Pearson correlation coefficient between the dominant genus and environmental variables. * $p < 0.05$, ** $p < 0.01$, *** $p < 0.001$.

## 4. Discussion

As the main production mode, the traditional pond polyculture system is a common practice in aquaculture [20]. It was reported that the high nutrient accumulation and low transfer ratio of nutrient output in ponds caused diverse biological activities in the water and sediments [20–22]. The present study demonstrated dynamic changes in water microbial composition and the driving environmental factors during a culture period in black carp polyculture ponds, a popular aquaculture mode in South China.

In the studied black carp polyculture ponds, the Shannon and Simpson diversity indices for aquatic bacteria increased during the mid-culture period (June, July, and August) in BP ($p < 0.05$), indicating that the bacterial diversity was in a dynamic succession process; the diversity decreased during the late culture period. The results were in accordance with those of a previous study [21], and the decreased species diversity in pond water may be due to the lesser nutrient availability in the late culture period.

For the microbial composition, Proteobacteria, Actinobacteria, and Bacteroidetes were the most abundant phyla in the water column of the black carp ponds during the culture period, consistent with previous studies in grass carp, crab, and shrimp ponds [22–24]. As the most abundant phyla in the aquaculture pond, Proteobacteria was reported to participate in various biochemical processes (for example, carbon, nitrogen cycling) in aquatic ecosys-

tems [25], and Actinobacteria are well-known bioactive natural product producers [26,27]. Phyla Cyanobacteria and Firmicutes, which reportedly to be more abundant in aquaculture pond [22], dominated the water column in the early (May) stage and middle (June, July, and August) stage, respectively. From LEfSe analysis on genus taxa, 13 abundant genera with significant difference were found between different months. One dominated genus (*Prochlorococcus*) was found in May and two genera (*Bacillus* and *Polynucleobacter*) in June and three genera (*Chryseobacterium*, *Novosphingobium* and *Acinetobacter*) in August, while seven genera (*Flavobacterium*, *Oscillospira*, *Ruminococcaceae*, *Agrobacterium*, *Comamonas*, *Janthinobacterium*, and *Rheinheimera*) were found in July. Suggesting more diverse bacterial communities existed in pond water in July and August, and these may be due to the high temperature, which was revealed to strongly influence the microbial communities in pond water [28,29]. Therefore, the microbial communities in the water column were divided into three groups. Samples collected in May clustered together, while samples collected in September and October clustered in another group, and samples collected in June, July, and August clustered together as a third group according to PCoA and UPGMA. These results suggest there are obvious seasonal differences in microbial composition in water column, which are consistent with previous studies [30].

From RDA analyses based on prevalent microbial taxa in this study, we found that nutrients such as TP, TN, $PO_4^{3-}$-P, and $NO_3^-$-N were significantly influence the microbial community in the water column, regardless of the size of the ponds. Dai et al. [22] has reported that nutrient levels all showed significant correlation with the microbial communities in pond water by studying aquaculture ponds from different areas in China. Moreover, T, pH, and DO were also positively correlated to the changes in microbial community composition, and they are reported to have strongly influence on bacterial growth and to contribute to ecosystem processes [31,32]. To further elucidate the driving environmental factors on special microbial taxa, we computed the Pearson correlation coefficient between the dominant genera and the environmental factors. The data revealed T to be the major environmental factor shaping the dominant microbial genera including *Prochlorococcus*, *Chryseobacterium*, *Acinetobacter*, *Rheinheimera*, *Polynucleobacter*, and *Janthinobacterium* in water column. Most of these genera mentioned were reported to be potential pathogenic bacteria [33], and *Flavobacterium* that was strongly correlated to SD also has been reported to be consistent of opportunistic pathogens [34–36], it will be an important challenge to manage the black carp pond by quickly monitoring the water temperature and SD for the future. As the more prevalent bacteria in pond water, *Polynucleobacter* was strongly negatively correlated to pH. It was reported that planktonic *Polynucleobacter* are abundant in many lakes and ponds, especially in acidic waters [37], and this is consistent with our result.

## 5. Conclusions

The current study was initiated to monitor the dynamics of bacterial communities and assess the influencing physicochemical parameters during a culture cycle in black carp polyculture ponds. Variation in microbial diversity was generally found in the water column during the culture period. Proteobacteria, Actinobacteria, and Bacteroidetes were the most abundant phyla, while the abundances of phyla Cyanobacteria, Firmicutes, and Bacteroidetes changed in different months. Moreover, 13 abundant genera with significant difference were found between different months. TP, TN, $PO_4^{3-}$-P, $NO_3^-$-N, T, pH, and DO significantly shaped the microbial community composition. While Pearson correlation coefficient showed that T, SD, and pH were strongly correlated to the 13 dominant genera. Considering some genera are potential pathogenic bacteria, we could manage the black carp pond by quickly monitoring the water temperature and SD in the future.

**Supplementary Materials:** The following are available online at https://www.mdpi.com/article/10.3390/w13213089/s1, Table S1: Environmental factors in BP (large areas) and SP (small areas) ponds during a culture period (Mean ± SD).

**Author Contributions:** The authors X.L. and L.L. contributed equally to this work and should be considered co-first authors. Field study and sampling were performed by X.L., L.L., Y.Z., T.Z., X.W. and D.Y.; Sample processing, Y.Z. (hydrochemistry), T.Z. and X.W. (bacteria); Formal data analysis, X.L. and L.L.; Writing—original draft preparation, all authors; Final text, X.L. and D.Y.; Writing—review and editing, all authors; Approval of the final text, all authors. All authors have read and agreed to the published version of the manuscript.

**Funding:** This study was funded by the Central Public-interest Scientific Institution Basal Research Fund, CAFS (2020XT1302), the National Key R&D Program of China (2019YFD0900603), and the China Agriculture Research System of MOF and MARA (CARS-46).

**Institutional Review Board Statement:** Not applicable.

**Informed Consent Statement:** Not applicable.

**Data Availability Statement:** Data are contained only within the article.

**Acknowledgments:** We thank the graduate students Xuge Wang and Degao Xu, for their help during sample collection. We also would like to thank three anonymous reviewers for their helpful comments.

**Conflicts of Interest:** The authors declare no conflict of interest.

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
