# Peer review of "Microbial Community Structure and Its Driving Environmental Factors in Black Carp (Mylopharyngodon piceus) Aquaculture Pond"

_water, doi:10.3390/w13213089_

Round 1

Reviewer 1 Report

The authors Li et al. monitored bacterial community structure in aquaculture ponds of black carp. Pond aquaculture has been a dominant approach in fish aquaculture worldwide, and it is critical learning the microorganisms living in the aquaculture ponds, such as their population dynamics during the fish cycle, ecosystem functions, their interaction with fish. These information are critical in understanding sustainable aquaculture and fish microbial ecology. The authors collected water samples from 6 aquaculture ponds from May to October, monitored the composition changes, and correlated their results with corresponding environmental factors. High throughput sequencing was applied to decipher the microbial community structure, and potential microbial functions (i.e., nitrogen transformation) were predicted via PICRUSt.  Major concerns from this reviewer:

  • Introduction: The authors need to review the current knowledge on microbial ecology on fish (carp) pond. It is important to know what the knowledge gap is and what this current work adds to the literature.
  • Need to tie in fish culture to the results. For instance, how microbial data interact with fish aquaculture? Any measurements for the aquaculture, such as fish population size, density, growth etc.? Also where are the metadata? The paper is about fish pond but the authors are missing the connection between microbes and aquaculture species.
  • What is the rational of separating the big vs. small ponds? Not surprisingly, the data showed no differences. May be able to merge the small and big ponds and treat them as replicates.
  • Why was the RDA analysis based on phylum level (section 2.5)? Need to provide further statistical support to groupings in PCoA analysis, in addition to visual observation. May consider ANOSIM to verify the groupings.   
  • Need to show the significant differences of predicted relative abundances of genes for Figure 6.
  • DNRA is an interesting observation and deserve further investigations. How were the DO measurements during the sampling period? In general the DO will be lower due to high temperature in summer, but the fish feeding also reduce DO to certain level.
  • Language improvement and proof-reading is strongly recommended.

Reviewer 2 Report

This manuscript appears interesting to show nitrogen cycling in aquaculture pond, but the presentation is poor to allow reviewers to understand its value.  The critical information in methodology for readers to understand its sampling and associated rational is missing. Also, environmental factors such as TP, TN, T, pH, DO etc are not provided to allow the readers to further understand what the conditions are. Discussion is also weak.

Critical questions for authors to answer:

  1. What is the DO level in ponds? Is DO level uniform from the pond surface to the bottom? As described, the pond depth is 3 meters. What is the position of DO probe in terms of depth of the pond from the surface to the bottom? How do we know that pond is fully aerobic or anoxic or anaerobic? I believe that ponds should be aerobic for Black Carp to survive. In this case, how could denitrification and dissimilatory nitrate reduction (DNR) occur?  In the method, authors mentioned several times about water surface. If sampling was conducted from water surface (assume with the highest dissolved oxygen level), how could denitrification and DNR occur?
  2. What is the water sampling depth in the ponds? Pls provide this information in M&M. In 2.1 and 2.2, water sampling were described twice with different methods? How are they different and why? It mentions that water samples were collected 2-3 sites. How to determine these 2-3 sites? What methodology was used to guide this 2-3 sites for water sampling?
  3. In the method, it says ‘The PCR products were separated by 2% agarose gel electrophoresis, and negative controls were always performed to make sure there was no contamination. The band with the correct size was excised and purified using a SanPrep  DNA Gel Extraction Kit’. What is the prupose for separating bands and excising bands? How are these related with DNA sequencing?
  4. Authors should provide all data about DO, PH, TP, TN, ammonium, nitrite, nitrate, phosphate etc in the supplementary document.
  5. Lines 163-168. Why are data single values? Are they average? If so, what data are used for average and what are the SD?
  6. Authors mentioned ‘various biogeochemical processes’ several times. If samples are only from water, what does ‘geo’ in the text mean?
  7. Line 255-256. There is no any data about fish feed. In this case, how could a nitrogen cycle be established in this manuscript when the pond is continuously receiving protein (from fish feed)?
  8. I am very curious how authors obtained conclusions on ‘The relative 299 abundances of N assimilation (53.37% ± 7.16%), N mineralization (7.85% ± 0.20%), and 300 DNRA (8.00% ± 1.99%) accounted for a substantial percentage of N cycling’. How could the assimilation be obtained? DNRA and N mineralization were obtained? Then what is the raining nitrogen in the nitrogen cycle? In addition, pls explain N assimilation, N mineralization clearly to show that what readers understand about these are what you are explaining.

Reviewer 3 Report

Authors of the present submission  did a good job laying out the need to balance N-input and output in aquaculture systems as an important approach to minimizing pollution associated with aquaculture. The manuscript represents a good effort but was confounded in my opinion by the addition and misinterpretations of the PICRUST data.The submission could benefit from major revisions prior to acceptance. Specific comments are below

Introduction

  1. authors failed to explain polyculture in the text at the first mention and throughout.
  2. Largely unknown is not a valid nor very valid reason for the study to be carried out. Please revise and add some depth.

METHODS

  1. Lines 95-97: what exactly does " the filters that were used were mixed " mean? Did you sequence the residue on the 0.45 and the 0.22 micron membranes separately or did you pool them? Clarify.
  2. Line 103: What is a PCR integrant?
  3. Line 124-125: Which version of the SILVA database did you use? Please state the version as they vary in reliability and accuracy of curated taxa.
  4. Why at the80% confidence leven an dnot 97%
  5. You used SILVA and RDP classifiers in assigning taxonomic information to the OTUs. why?
  6. PICRUST  seems unneccessary for this type of generated data. The information from PICRUST would have been better served if it came from  a metagenome. As is you are using 16s marker gene to get at some information from PICRUST and KEGG that arent very intuitive and not appropriate.
  7. Line 145: Explain how you were able to arrive at 36 samples.
  8. Line 149-152: So Chao1 and PD showed no significant differences compared to shannon and simpson indices. Why is that, and perhaps just pick one or two of these indices that speaks to richness and evenness of the data and not all 4 .
  9. Lines 201-228:  You cannot assume nor infer POTENTIAL N- metabolism data or expression level from Miseq DNA data!! This is not expression data as no RNA nor cDNA were generated before use. Just because you used PICRUST OR KEGG doesnt make this the kind of data. This is wrong and this section need to be removed form the mauscript. Carry out qPCR on these genes if needed but without that kind of data this is an absolute error. You essentially measured the relative abundances  from the PICRUST based solely on the 16 marker gene and using the phrase, expression level. This has no functional value.
  10. lINES 243-260: The attribution of certain microbial functions to rather specific taxa is a bit problematic.
  11. Lines 247-249: This lines do not make sense. Did you find probiotic genes? This is the first mention of this. If it was not searched fro, then omit.
  12. Lines 262-280: Remove this section as what was actually measured cannot be equated to expression levels. Same applies to  lines 280-289

Round 2

Reviewer 1 Report

The authors have fully addressed the concerns from this reviewer.

Reviewer 2 Report

accepted 

Reviewer 3 Report

Glad the authors agreed with comments and reviews. The manuscript reads much better